

# Typical synoptic situations and their impacts on the wintertime air pollution in the Guanzhong basin, China

N. Bei[1,2], G. Li[2,3], R. Huang[2,4], J. Cao[2,3], N. Meng[1], T. Feng[2,3], S. Liu[2,3], T. Zhang[2,3], Q. Zhang[5], and L. T. Molina[6]

[1]School of Human Settlements and Civil Engineering, Xi'an Jiaotong University, Xi'an, China
[2]Key Laboratory of Aerosol Chemistry and Physics, Institute of Earth Environment, Chinese Academy of Sciences, Xi'an, China
[3]State Key Laboratory of Loess and Quaternary Geology, Institute of Earth Environment, Chinese Academy of Sciences, Xi'an, China
[4]Laboratory of Atmospheric Chemistry, Paul Scherrer Institute (PSI), 5232 Villigen, Switzerland
[5]Department of Environmental Sciences and Engineering, Tsinghua University, Beijing, China
[6]Molina Center for the Energy and the Environment, La Jolla, CA, USA

Received: 12 September 2015 – Accepted: 9 December 2015 – Published: 15 January 2016

Correspondence to: N. Bei (bei.naifang@mail.xjtu.edu.cn) and G. Li (ligh@ieecas.cn)

Published by Copernicus Publications on behalf of the European Geosciences Union.

**ACPD**

doi:10.5194/acp-2015-710

**Typical synoptic situations and their impacts on air pollution**

N. Bei et al.

# Abstract

Rapid industrialization and urbanization have caused severe air pollution in the Guanzhong basin, northwestern China with heavy haze events occurring frequently in recent winters. Using the NCEP reanalysis data, the large scale synoptic situations influencing the Guanzhong basin during wintertime of 2013 are categorized into six types to evaluate the contribution of synoptic situations to the air pollution, including "north-low", "southwest-trough", "southeast-high", "transition", "southeast-trough", and "inland-high". The FLEXPART model has been utilized to demonstrate the corresponding pollutant transport patterns for the typical synoptic situations in the basin. Except "southwest-trough" and "southeast-high" (defined as favorable synoptic situations), the rest four synoptic conditions (defined as unfavorable synoptic situations) generally facilitate the accumulation of air pollutants, causing heavy air pollution in the basin. In association with the measurement of $PM_{2.5}$ (particulate matter with aerodynamic diameter less than 2.5 μm) in the basin, the unfavorable synoptic situations correspond to high $PM_{2.5}$ mass concentrations or poor air quality and vice versa. The same analysis has also been applied to winters of 2008–2012, which shows that the basin was mainly influenced by the unfavorable synoptic situations during wintertime leading to poor air quality. The WRF-CHEM model has further been applied to simulate the selected six days representing the typical synoptic situations during the wintertime of 2013, and the results generally show a good consistence between the modeled distributions and variations of $PM_{2.5}$ and the corresponding synoptic situations, demonstrating reasonable classification for the synoptic situations in the basin. Detailed meteorological conditions, such as temperature inversion, low-level horizontal wind speed, vertical wind velocity, and convergence all contribute to heavy air pollution events in the basin under unfavorable synoptic conditions. Considering the proportion of occurrence of unfavorable synoptic situations during wintertime, reduction of emissions is the optimum approach to mitigate the air pollution in the Guanzhong basin.

Discussion Paper | Discussion Paper | Discussion Paper | Discussion Paper |

**ACPD**

doi:10.5194/acp-2015-710

**Typical synoptic situations and their impacts on air pollution**

N. Bei et al.

# 1 Introduction

Elevated atmospheric pollutants, such as particulate matter (PM) and ozone ($O_3$), exert deleterious impacts on human health and environment (e.g., Penner et al., 2001; Pope and Dockery, 2006; J. Zhang et al., 2010). Over the past three decades, with tremendous economic growth in China, rapid industrialization and urbanization have caused severe air pollution, as reflected in the heavy haze event that often occurs in the north of China, particularly during wintertime (e.g., Chan and Yao, 2008; Fang et al., 2009; Gao et al., 2011; Liu et al., 2013; Zhao et al., 2013; Huang et al., 2014; Fu et al., 2014; Guo et al., 2014; Han et al., 2014; Zhang et al., 2015; Yang et al., 2015). Guanzhong basin is located in the northwest of China, nestled between the Qinling Mountains in the south and the Loess Plateau in the north. The unique topography facilitates the accumulation of air pollutants, and with the rapid increasing industries and city expansions, heavy air pollution frequently attacks the basin (e.g., Cao et al., 2009; Sheng et al., 2011).

Numerous studies have demonstrated that the meteorological conditions play an important role in the formation, transformation, diffusion, transport, and removal of the atmospheric pollutants (e.g., Seaman, 2000; Solomon et al., 2000; de Foy et al., 2005, 2006; Bei et al., 2008, 2010, 2012, 2013). If the emissions of pollutants remain invariable, transformations in the chemical state of the atmosphere are principally determined by the meteorological conditions. Recent advances in understanding the role of the meteorological conditions in the air pollution formation in China have mainly concentrated on the regions of Beijing-Tianjin-Hebei, the Pearl River Delta, and the Yangtze River Delta (e.g., Wu et al., 2008, 2013; Wang et al., 2009; Q. H. Zhang et al., 2010; Gao et al., 2011; Zhang et al., 2012; L. Wang et al., 2014; H. Wang et al., 2014; Zhang et al., 2015). Wang et al. (2009) have shown that the $O_3$ decrease at a Beijing rural site during the 2008 Olympics is attributed to the favorable meteorological condition in comparison with the same period in 2006 and 2007. Q. H. Zhang et al. (2010) have proposed that, during the 2008 Olympics, the atmospheric visibility improvements

are likely caused by the decrease of atmospheric relative humidity compared to the same period in the previous 5 years. Using a coupled meteorology-chemistry model, Gao et al. (2011) have further pointed out that meteorological conditions are as important as emission controls in reducing aerosol concentrations in Beijing during the
2008 Olympics. Wu et al. (2008) have performed analysis of the typical haze and clean weather processes over the Pearl River Delta in 2004 and 2005, and found that the regional haze formation is highly correlated to the regional calm wind process while the cleaning process is influenced by the strong advection transport. Additionally, Wu et al. (2013) have classified two typical weather conditions associated with poor air
quality over the Pearl River Delta, including the warm period before a cold front and the subsidence period controlled by a tropical cyclone during two intensive observations in 2004 and 2006. L. Wang et al. (2014) have demonstrated that recirculation and regional transport, along with the poorest diffusion conditions and high humidity favorable for hygroscopic growth of secondary aerosols, caused the extremely high levels
of $PM_{2.5}$ in Beijing during January 2013. Zhang et al. (2015) have also suggested that the weak transport/diffusion was an important factor for the haze occurrences.

   The circulation-based classification is an approach to identify synoptic weather categories through determining the circulation types from sea level pressure, geopotential height, or wind fields (Huth et al., 2008). Since the meteorological fields that affect the
air quality are generally closely interrelated and strongly controlled by the synoptic-scale circulation, the circulation classification has been extensively used in environmental studies, especially in the middle and high latitude regions where local weather conditions are chiefly determined by the day-to-day synoptic circulation variability (e.g., Jacobeit, 2010; Huth et al., 2008). For example, Bei et al. (2013) have classified the typ-
ical synoptic situations and the associated plume transport patterns in the US-Mexico border region along the Pacific Ocean, and found that the plume transport directions are generally consistent with the prevailing wind directions on 850 hPa. However, only a few studies have been performed to investigate the synoptic weather classification in China (Huth et al., 2008). Cheng et al. (2001) have demonstrated that high $O_3$

Discussion Paper | Discussion Paper | Discussion Paper | Discussion Paper |

**[ACPD](doi:10.5194/acp-2015-710)**

doi:10.5194/acp-2015-710

**Typical synoptic situations and their impacts on air pollution**

N. Bei et al.

concentrations at Taiwan are related to anticyclonic synoptic systems and a tropical low-pressure system moving from the south of Taiwan. Recent studies have also indicated that the developing and different parts of an anticyclonic system play an important role in regulating air quality (Chen et al., 2008; Wei et al., 2011). However, the above-mentioned studies are all based on case studies during a short period using subjective procedure. Zhang et al. (2012) have verified the relationship between surface circulation pattern and air quality in Beijing and the surrounding areas over 10 years using a synoptic approach based on an objective classification procedure (Philipp et al., 2010). Results have demonstrated that significant differences exist in the local meteorology and footprints of 48 h backward trajectories among various circulation types, and synoptic-scale circulations are the principal drivers of day-to-day variations in pollutant concentrations over Beijing and surrounding areas during the emission control period.

Previous studies have examined the composition, characteristics, and sources of the atmospheric pollutants in the Guanzhong basin (e.g. Cao et al., 2009, 2012; Shen et al., 2010, 2011). However, few studies have been performed to comprehensively explore the relationship between air pollution and the meteorological conditions at both synoptic and local scales in this area. Therefore, it is imperative to examine the role of the specific meteorological conditions in the formation of heavy air pollution in this area to support design and implementation of emission control strategies.

The purpose of the present study was to categorize the large-scale synoptic weather systems that impact the Guanzhong basin in the winter along with the measurements of pollutants in the basin using a subjective classification procedure, which determines the circulation types from the geopotential height and wind fields on 850 hPa. The air quality situations associated with various synoptic situations are simulated using the WRF-CHEM model developed by Li et al. (2010, 2011a; 2011b), to evaluate the contributions of the particular meteorological conditions to the severe air pollution. The models and methodology used in this study are introduced in Sect. 2. The main results are presented in Sect. 3. Conclusions and discussions are given in Sect. 4.

**ACPD**

doi:10.5194/acp-2015-710

**Typical synoptic situations and their impacts on air pollution**

N. Bei et al.

## 2   Data, models, and methodology

The National Centers for Environmental Prediction (NCEP) final operational global gridded analysis (FNL) ($1° \times 1°$) is used to categorize the large-scale synoptic weather systems influencing the Gunazhong basin during the period from 2008 to 2013 through the subjective procedure. The geopotential height and wind fields on 850 hPa are applied to identify the synoptic situations that affect the plume transport patterns in the basin.

Continuous daily $PM_{2.5}$ measurements have been performed at the Institute of Earth Environment, Chinese Academy of Sciences (IEECAS) in Xi'an, China since 2003. Additionally, since January 2013, the China's Ministry of Environmental Protection (China MEP) has commenced to release the real-time hourly concentrations of $PM_{2.5}$. Total 33 monitoring sites are distributed in the Guanzhong basin (Fig. 1b). The daily $PM_{2.5}$ measurement at IEECAS site from 2008 to 2012 and the hourly $PM_{2.5}$ measurement released by China MEP from 2013 to 2014 are used to validate the categorized synoptic situations influencing the basin.

In order to analyze the corresponding pollutant transport patterns under the typical categorized synoptic situations, The FLEXPART model is employed to calculate the forward Lagrangian particle dispersion (Stohl et al., 1998; Fast and Easter, 2006), which is driven by the output from the WRF model (Skamarock et al., 2008). The FLEXPART model is set-up with releases of 6000 computational particles within a grid cell of $10 km \times 10 km \times 0.02 km$ centered at Xi'An urban area in the morning. Tracer particles are released continuously from 04:00 to 10:00 BJT (Beijing Time) of the day, and traced until 04:00 BJT of next day. For the convenience, all the time used hereafter is BJT. The WRF model adopts one grid with horizontal resolution of 3 km and 35 sigma levels in the vertical direction. The grid cells used for the domain are $201 \times 201$ (Fig. 1a). The selected six days, representing six categorized typical synoptic situations of the Guanzhong basin during wintertime of 2013, are simulated. They are initialized at 20:00 BJT on each day and integrated for 36 h. The NCEP FNL analysis data ($1° \times 1°$) is used to produce the initial and boundary conditions for the WRF model. The physi-

**[ACPD](doi:10.5194/acp-2015-710)**

doi:10.5194/acp-2015-710

**Typical synoptic situations and their impacts on air pollution**

N. Bei et al.

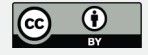

cal process parameterization schemes used in simulations included the Grell-Devenyi ensemble scheme for cumulus scheme (Grell and Devenyi, 2002), the WRF Single Moment (WSM) three-class microphysics (Hong et al., 2004), and Mellor–Yamada–Janjic (MYJ) TKE scheme (Janjic, 2002) for the PBL processes.

5    The WRF-CHEM model has been used to further simulate the selected six days representing the typical categorized synoptic situations and to verify the particular meteorological conditions during the severe air pollution events in the Guanzhong basin. A specific version of the WRF-CHEM model (Grell et al., 2005) is used in the present study, which was developed by Li et al. (2010, 2011a, b, 2012) at the Molina Center for Energy and the Environment, with a new flexible gas phase chemical module and the CMAQ (version 4.6) aerosol module developed by US EPA (Binkowski and Roselle, 2003). The inorganic aerosols are simulated in the WRF-CHEM model using ISORROPIA ("equilibrium" in Greek, here referred to as an improved thermodynamic equilibrium aerosol model) Version 1.7 (http://nenes.eas.gatech.edu/ISORROPIA/). The secondary organic aerosols (SOA) formation is simulated using a non-traditional SOA model including the volatility basis-set modeling method in which primary organic components are assumed to be semi-volatile and photochemically reactive and are distributed in logarithmically spaced volatility bins (Li et al., 2011a). Detailed description of the WRF-CHEM model can be found in Li et al. (2010, 2011a, b, 2012). The meteorological setup in the WRF-CHEM model simulations is same as those in the WRF model, except that the spin-up time of the WRF-CHEM model is one day. The chemical initial and boundary conditions for the WRF-CHEM model simulations are interpolated from the 6 h output of a global chemical transport model for $O_3$ and related chemical tracers (MOZART). The anthropogenic emission inventory (EI) developed by Zhang et al. (2009) is used in the study, including contributions from agriculture, industry, power, residential and transportation sources. The MEGAN model developed by Guenther et al. (2006) is used to calculate on-line biogenic emissions.

**[ACPD](doi:10.5194/acp-2015-710)**

doi:10.5194/acp-2015-710

**Typical synoptic situations and their impacts on air pollution**

N. Bei et al.

# 3   Results

NCEP-FNL reanalysis data, the model output from the FLEXPART model, and the PM$_{2.5}$ measurements in the Guanzhong basin are used to explore the typical meteorological synoptic situations and the corresponding plume transport patterns. Using the model output from the WRF-CHEM model, we have further investigated the local meteorological conditions, including the PBL height, low level wind speed, inversion layer, low level convergence and divergence, and vertical wind velocity, and their potential impacts on the air pollution formation process in the basin.

## 3.1   Classification of the typical synoptic situations and the corresponding pollutant transport patterns

Based on the NCEP-FNL reanalysis data, we have first performed the analysis of the synoptic situations during the wintertime of 2013 using the 850 hPa wind and geopotential height fields. Here the wintertime is defined as December of the year to February of the next year. Six typical synoptic situations are categorized, including "north-low", "southwest-trough", "southeast-high", "transition", and "inland-high". The detail dates are shown in Table 1. The percentages of the above-mentioned six types are 17.8, 14.4, 4.4, 12.2, 5.6, and 45.6 %, respectively, indicating that the "inland-high" is the dominant wintertime synoptic situation influencing the Guanzhong basin.

Figure 2 shows 850 hPa winds and geopotential heights at 08:00 BJT for the selected 6 days representing the six categorized typical synoptic situations, respectively, including (1) 16 February 2014 ("north-low"), (2) 19 January 2014 ("southwest-trough"), (3) 26 December 2013 ("southeast-high"), (4) 2 December 2013 ("transition"), (5) 23 January 2014 ("southeast-trough"), and (6) 23 December 2013 ("inland-high"). In case of the "north-low" (Fig. 2a), the Guanzhong basin is generally located in the north of the low on 850 hPa and the weak east wind is prevalent aloft. Due to the blocking of the specific topography (Fig. 1a), the convergence or stagnant conditions are frequently formed, which is not favorable for the dispersion of air pollutants. However, heavy air

Discussion Paper | Discussion Paper | Discussion Paper | Discussion Paper | Discussion Paper

**ACPD**

doi:10.5194/acp-2015-710

**Typical synoptic situations and their impacts on air pollution**

N. Bei et al.

pollution might not emerge in the basin due to the occurrence of precipitation, which is caused by the favorable dynamical conditions that can efficiently clean up the pollutants. For the category of "southwest-trough" (Fig. 2b), the basin is located in the southwest of the trough on 850 hPa and the northwest wind is prevailing over the basin. The cold and dry air from the northwest effectively evacuates the air pollutants formed in the basin and also brings blue sky for the basin. For the category of "southeast-high" (Fig. 2c), the high of 850 hPa in the northwest of the basin originates the prevalent northeasterly or northwesterly winds, transporting the pollutants outside of the basin and remarkably improve the air quality. When the basin is situated in the transition area between the trough in the north and the high in the south on 850 hPa, the synoptic situation is defined as the category of "transition" (Fig. 2d). The prevailing winds over the basin are generally westerly, but the west wind is significantly attenuated by the topography in the basin and often transformed to be calm or disordered. The atmospheric pollutants are subject to be conveyed to the east of the basin, but more likely to be trapped in the basin, causing heavy air pollutions. Additionally, in the condition of "transition", the meteorological fields in the basin are also coordinately adjusted with the development of the trough in the north and the high in the south on 850 hPa. Under the condition of "southeast-trough" (Fig. 2e), the basin is affected by the trough on 850 hPa in the northwest, and the southwest wind is dominant aloft. The weak south winds and convergence formed in front of the trough tend to withhold the air pollutants in the basin, significantly deteriorating the air quality. For the category of "inland-high" (Fig. 2f), the basin is controlled by the inland high on 850 hPa, and the prevailing wind is varied over the basin, depending on the detailed location of the high. The situation of weak winds, subsidence, and the stable stratification facilitates the accumulation of atmospheric pollutants, often causing severe air pollutions in the basin.

Figure 3 displays the 24 h plume transport patterns initialized from 04:00 BJT on the abovementioned 6 representative days. The particles released in the morning in the urban area of Xi'an are generally transported within the planetary boundary layer (PBL). Apparently, only in case of "southwest-trough" and "southeast-high", the particles can

**ACPD**

doi:10.5194/acp-2015-710

**Typical synoptic situations and their impacts on air pollution**

N. Bei et al.

**ACPD**

doi:10.5194/acp-2015-710

**Typical synoptic situations and their impacts on air pollution**

N. Bei et al.

be transported outside of the basin and the improvement of air quality in the basin is anticipated. For the rest four synoptic categories, most of released particles ramble in the basin, indicating buildup of the air pollutants.

The above analyses demonstrate that only two kinds of synoptic conditions ("southwest-trough" and "southeast-high", defined as favorable synoptic situations hereafter) disperse pollutants efficiently and engender the good air quality in the Guanzhong basin. The rest four kinds of synoptic situations ("north-low", "transition", "southeast-trough", and "inland-high", defined as unfavorable synoptic situations hereafter) are generally favorable for the accumulation of the air pollutants either in horizontal or vertical directions, except in case of the "north-low" with strong winds or precipitation or the "southeast-trough" with strong vertical mixings.

Figure 4 displays the average diurnal cycle of observed $PM_{2.5}$ mass concentrations at 33 monitoring sites in the Guanzhong basin under the six synoptic categories during the wintertime of 2013. Consistently, the favorable synoptic situations correspond to low $PM_{2.5}$ mass concentrations or relatively good air quality and vice versa. Under the unfavorable synoptic situations, the observed average $PM_{2.5}$ mass concentrations generally range from 150 to $250\,\mu g\,m^{-3}$, showing that the basin has experienced heavy air pollution. The $PM_{2.5}$ mass concentrations in case of "north-low" are lower than those under the rest three unfavorable synoptic situations, which is caused by the possible occurrence of precipitation in the condition of "north-low". For example, the synoptic patterns on 4 and 5 February 2014 are categorized to "north-low", but the observed average $PM_{2.5}$ mass concentrations are less than $90\,\mu g\,m^{-3}$ because of the precipitation washout on these two days. Although the favorable synoptic situations facilitate the evacuation of air pollutants in the basin, the observed average $PM_{2.5}$ mass concentrations still exceed $35\,\mu g\,m^{-3}$, indicating that the air quality in the basin barely reaches the excellent level. It should be noted that the exceptional days exist beyond the six synoptic situations, indicating the complexity of atmospheric circulations.

With the same method as used in 2013, we have further classified the large-scale synoptic situations of the wintertime in the Guanzhong basin for the period from 2008 to

ACPD

doi:10.5194/acp-2015-710

**Typical synoptic situations and their impacts on air pollution**

N. Bei et al.

2012. The above-mentioned six typical synoptic situations influencing the Guanzhong basin during the wintertime from 2008 to 2012 are summarized in Table 2. Figure 5 displays the daily mean $PM_{2.5}$ mass concentration averaged during the six typical synoptic situations from 2008 to 2012 at the IEECAS site. The percentage of total unfavorable synoptic situations during 2008 to 2012 is about 85 %, and corresponding daily $PM_{2.5}$ mass concentrations exceed 200 µg m$^{-3}$ (Fig. 5), indicating the significant contribution from the large-scale meteorological conditions to the poor air quality in the basin. The "inland-high" dominates the synoptic situation in association with the poor air quality in the basin, with the contribution of around 43 %. The favorable situations constitute about 15 % of the synoptic situation in the basin, which is anticipated to empty the basin and significantly improve the air quality. However, the observed daily $PM_{2.5}$ mass concentrations during the favorable situations still exceed 75 µg m$^{-3}$ and fail to reach the good level, indicating the massive local emissions of pollutants and considerable contributions of background dust transport from Loess plateau in the north.

## 3.2 Local meteorological conditions on the selected 6 days and their impact on the air quality

The results of the FLEXPART model only explain the direct impact of the meteorological fields on the plume transport process since chemical processes are not considered in the model. The WRF-CHEM model is therefore used to simulate the air quality in the Guanzhong basin on the selected six days corresponding to the above-mentioned six kinds of typical synoptic situations.

Figure 6 provides the vertical distributions of temperature, wind vectors, and PBL height through Xi'An along the east–west direction at 09:00 and 15:00 BJT on the selected six days, in order to investigate the vertical atmospheric characteristics under the six typical synoptic situations. The vertical section shows the depth of the basin is around 1 km, indicating that the local terrain (Loess Plateau and Qinling Mountains) has important impacts on the low-level wind fields inside the basin. Under unfavorable

Discussion Paper | Discussion Paper | Discussion Paper | Discussion Paper

synoptic situations, the winds inside the basin are remarkably attenuated due to the influence of the terrain, favorable for trapping the air pollutants formed in the basin. In addition, at 09:00 BJT, temperature inversions in case of unfavorable synoptic situations also impede the development of PBL, decreasing the diffusion of air pollutants in the vertical direction. At 15:00 BJT, the weak winds do not boost the PBL development due to lack of the wind shear inside the basin and the thermal impact dominates the PBL height. The PBL on 2 December 2013 and 23 January 2014 is higher than that on 16 February 2014 and 23 December 2013 due to the low-level temperature discrepancy, and the impact of the urban heat island on the PBL height is also obvious. Therefore, the unfavorable synoptic situations are prone to trap the pollutants inside the basin due to inefficient horizontal transportation and impeded vertical diffusion, leading to the heavy air pollution in the basin. In case of favorable synoptic situations, the strong horizontal wind (19 January 2014) or active vertical motion (26 December 2013) efficiently diffuse the pollutants in the horizontal or vertical directions and the high PBL also expedites the vertical exchange of air pollutants in the basin, so the good air quality is expected.

To investigate the detailed local meteorological conditions over the Guanzhong basin on the above-mentioned six days, we have further analyzed the low-level (below 850 hPa) vertical motion, divergence, and horizontal wind speed averaged over the Guanzhong basin (the averaged domain indicated in Fig. 1). Figure 7 shows the time-evolutions of the area averaged low-level vertical velocity, divergence, and wind speed over the basin on the selected 6 days. In general, under favorable synoptic situations, the divergence exists in the low-level atmosphere inside the basin, leading to the strong downward motion and outflowing of pollutants from the basin. In addition, the occurrence of strong horizontal winds also speeds up the evacuation of pollutants, such as on 19 January 2014, the average wind speed is around $8\,\mathrm{m\,s^{-1}}$. Under the unfavorable conditions, except on 2 December 2013, the weak convergence leads to slow upward motions, which withholds the pollutants inside the basin, and the weak horizontal winds also inefficiently disperse the pollutants, i.e., the horizontal wind speed is

Discussion Paper | Discussion Paper | Discussion Paper | Discussion Paper |

ACPD

doi:10.5194/acp-2015-710

**Typical synoptic situations and their impacts on air pollution**

N. Bei et al.

about $2\,\mathrm{m\,s^{-1}}$ on 23 December 2013. On 2 December 2013 ("transition"), the basin is influenced by the trough in the north and the high in the south on 850 hPa, and the variation of meteorological conditions inside the basin are determined by the development of the trough and the high. From the early morning to the noontime, the airflow inside the basin varies from convergence to divergence and the wind gets stronger, indicating the deepening of the trough in the north and the possible evacuation of pollutants from the basin. In general, the synoptic pattern influencing the basin experience the transition from "inland-high" to "southwest-trough", so the pollutants accumulated in the basin in the morning have potentials to be transported outside of the basin in the afternoon/evening, depending on the deepening of the trough in the north on 850 hPa.

Figure 8 presents the observed and simulated spatial distributions of near-surface $PM_{2.5}$ mass concentrations along with the modeled wind fields in the Guanzhong basin at 09:00 and 15:00 BJT on the selected 6 days. The calculated patterns of $PM_{2.5}$ mass concentrations are generally consistent with the observation over the ambient monitoring sites on those days. Under the favorable situations, the strong north or northwest winds have commenced to evacuate the air pollutants accumulated during nighttime at 09:00 BJT (Fig. 8b and c), and the whole basin becomes clean at 15:00 BJT. In case of unfavorable situations, the near-surface winds in the basin are weak or calm and frequently disordered, which facilitates the accumulation of air pollutants, causing heavy air pollution. The modeled and observed $PM_{2.5}$ mass concentrations exceed $150\,\mu\mathrm{g\,m^{-3}}$ at most of monitoring sites. Particularly, on 23 December 2013 ("inland-high"), the basin experienced severe air pollution with the $PM_{2.5}$ mass concentrations exceeding 250 or even $500\,\mu\mathrm{g\,m^{-3}}$ at monitoring sites. On January 23, 2014 ("southeast-trough"), the near-surface south winds over Qinling Mountains are not weak, but, apparently, the warm and humid air from the south does not significantly influence the wind fields in the basin. Unfortunately, the south winds over the Qinling Mountains carry the warm air aloft the basin, causing the temperature inversion and further hindering the diffusion of air pollutants in the vertical direction (Fig. 6e). So in the condition of "southeast-trough", the whole basin seems to be sealed and is

**ACPD**

doi:10.5194/acp-2015-710

**Typical synoptic situations and their impacts on air pollution**

N. Bei et al.

Discussion Paper | Discussion Paper | Discussion Paper | Discussion Paper

often severely polluted, i.e., the 5 year average filter measured PM$_{2.5}$ mass concentration exceeds 300 µg m$^{-3}$ at the IEECAS site (Fig. 5). Figure 9 provides the comparison of observed and predicted diurnal profiles of the PM$_{2.5}$ mass concentrations averaged over the monitoring sites in the Guanzhong basin on the selected six days.

The WRF-CHEM model generally captures well the observed diurnal variations of the PM$_{2.5}$ mass concentrations, but it often underestimates the observation in the early morning and overestimates during rush hours. Under favorable synoptic situations, the average PM$_{2.5}$ mass concentrations over monitoring sites are significantly decreased from the early morning to the late afternoon, and the air quality can reach the good level

during daytime. However, in the conditions of unfavorable synoptic situations, the average PM$_{2.5}$ mass concentrations are less decreased or even increased in the afternoon when the PBL well develops. The unfavorable synoptic situations generally induce the stagnant circumstances, retaining the air pollutants inside the basin. In the afternoon with the peak of sunlight, the elevated air pollutants, including the precursors of sec-

ondary aerosols, cause the rapid formation of secondary aerosols, such as nitrate and SOA, compensating the decrease of PM$_{2.5}$ mass concentrations due to the development of the PBL. Furthermore, high levels of aerosols in the low-level atmosphere also scatter the solar radiation and reduce the surface temperature, suppressing the development of the PBL. So the formation of secondary aerosols and the aerosol radiation

feedback likely lead to the high level PM$_{2.5}$ mass concentration in the afternoon in case of unfavorable synoptic situations. Figure 10 displays a scatterplot of the measured vs. modeled daily mean mass concentration of major aerosol constituents at IEECAS site on the selected six days. The filter measured organic carbon is scaled by a factor of 1.8 to compare with the simulated organic aerosol (Carlton et al., 2010). The

WRF-CHEM model performs reasonably in simulating daily mean sulfate, ammonium, organic aerosols and elemental carbon. However, the model consistently overestimates the observed ammonium aerosol mass concentration. The filter measurements show that the PM$_{2.5}$ in the basin contains abundant potassium, sodium, and calcium ions which are originated from biomass burning or dust and able to preferentially replace

Discussion Paper | Discussion Paper | Discussion Paper | Discussion Paper | Discussion Paper |

**ACPD**

doi:10.5194/acp-2015-710

**Typical synoptic situations and their impacts on air pollution**

N. Bei et al.

the ammonium ions. In the WRF-CHEM simulations, the contributions of potassium, sodium, and calcium ions are not considered due to lack of available emissions inventories, so when the ammonia is sufficient in the atmosphere, the overestimation of ammonium aerosols can be explained. In general, the simulated $PM_{2.5}$ patterns and variations on the selected six days are well consistent with the corresponding synoptic situations.

## 4   Conclusions and discussions

In the present study, the typical synoptic situations influencing the Guanzhong basin during wintertime have been investigated to evaluate their potential impacts on the air quality in the basin through using NCEP reanalysis data, aerosol measurements, and simulations by the FLEXPART, WRF and WRF-CHEM models. The results show that the synoptic situations significantly contribute to the air pollution in the basin during wintertime.

Based on the NCEP reanalysis data, the large-scale synoptic situations influencing the Guanzhong basin during the wintertime of 2013 are categorized into six types, including "north-low", "southwest-trough", "southeast-high", "transition", "southeast-trough", and "inland-high". The FLEXPART trajectory model has been utilized to examine the corresponding pollutant transport patterns for the typical synoptic situations in the basin. The pollutants are transported outside of the basin in case of "southwest-trough" and "southeast-high", which are defined as favorable synoptic situations. The rest four types of synoptic conditions, defined as unfavorable synoptic situations, are subject to pollutant accumulation in the basin, causing heavy air pollution. In association with the $PM_{2.5}$ measurements released by the China MEP, the favorable synoptic situations efficiently decrease $PM_{2.5}$ mass concentrations or significantly improve the air quality in the basin and vice versa.

The analysis of the large-scale synoptic situations of the wintertime during 2008 to 2012 shows that unfavorable synoptic situations constitute about 85 % of the winter

Discussion Paper | Discussion Paper | Discussion Paper | Discussion Paper | Discussion Paper |

**ACPD**

doi:10.5194/acp-2015-710

**Typical synoptic situations and their impacts on air pollution**

N. Bei et al.

days, indicating the significant contribution from the large-scale meteorological conditions to the poor air quality in the Guanzhong basin. In addition, the percentage of "inland-high" is around 42 %, which is the most popular synoptic situation associated with the poor air quality in the basin.

The WRF-CHEM model has been further used to simulate the selected six days representing the typical synoptic situations during the wintertime of 2013, and the results shows that the modeled $PM_{2.5}$ distribution and variations are generally consistent well with the corresponding synoptic conditions, which demonstrates the critical role of the synoptic meteorological conditions in air pollution events in the basin. The WRF-CHEM model simulations also indicate the reasonable classification for the synoptic situations in the basin. In addition, detailed meteorological conditions, including temperature inversion, low-level horizontal wind speed, vertical wind velocity, and convergence are also analyzed for the selected days. Under unfavorable synoptic situations, temperature inversion, weak low-level wind and convergence do not facilitate the dispersion of pollutants in the basin. While in the favorable synoptic situations, low-level divergence, caused by strong horizontal winds or active vertical motions, efficiently evacuate air pollutants in the basin.

During wintertime, 5 year filter $PM_{2.5}$ measurement from 2008 to 2012 and the $PM_{2.5}$ measurement released by China MEP in 2013 and 2014 all show that the Guanzhong basin has experienced heavy air pollution. Even under favorable synoptic situations, the observed $PM_{2.5}$ mass concentrations have barely reached the excellent level due to massive local emissions of air pollutants and the background dust transportation from Loess Plateau. Hence, considering the proportion of occurrence of unfavorable synoptic situations during wintertime, reduction of emissions is a feasible method to reduce the air pollution in the Guanzhong basin.

Given that the synoptic situations categorized are made at 850 hPa that influence the Guanzhong basin, potential uncertainties still exist in the classification results. More quantitative studies are needed in the future to improve the synoptic situation classification. Further, the analysis using local meteorological observations on ground surfaces

**[ACPD](doi:10.5194/acp-2015-710)**

doi:10.5194/acp-2015-710

**Typical synoptic situations and their impacts on air pollution**

N. Bei et al.

and inside the PBL is also imperative to investigate the role of the local meteorological conditions in the severe pollution events.

*Acknowledgements.* Naifang Bei is supported by the National Natural Science Foundation of China (no. 41275101) and the Fundamental Research Funds for the Central Universities of China. Guohui Li is supported by "Hundred Talents Program" of the Chinese Academy of Sciences and the National Natural Science Foundation of China (no. 41275153).

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

**Table 1.** Synoptic categories influencing the Guanzhong basin during the wintertime of 2013.

| Categories | Date[*] | Sum | Percentage (%) |
|---|---|---|---|
| North-low | 20131217 20131208 20140110 20140111 20140102 20140106 20140204 20140205 20140206 20140215 20140216 20140217 20140224 20140225 20140228 20140208 | 16 | 17.8 |
| Southwest-trough | 20131201 20131205 20131209 20131211 20131212 20131215 20131231 20140103 20140104 20140119 20140120 20140108 20140112 | 13 | 14.4 |
| Southeast-high | 20131210 20131226 20140107 20140227 | 4 | 4.4 |
| Transition | 20131202 20131207 20131214 20131230 20140101 20140116 20140127 20140130 20140201 20140202 20140207 | 11 | 12.2 |
| Southeast-trough | 20131204 20140123 20140129 20140131 20140226 | 5 | 5.6 |
| Inland-high | 20131203 20131206 20131213 20131219 20131221 20131222 20131223 20131224 20131216 20131225 20131218 20131220 20131227 20131229 20131228 20140105 20140113 20140118 20140122 20140125 20140115 20140121 20140126 20140128 20140109 20140114 20140117 20140124 20140214 20140219 20140211 20140210 20140213 20140220 20140221 20140222 20140223 20140218 20140212 20140209 20140203 | 41 | 45.6 |

[*] The format of date is YYYYMMDD, in which YYYY, MM, and DD represent year, month, day, respectively.

Discussion Paper | Discussion Paper | Discussion Paper | Discussion Paper

Discussion Paper | Discussion Paper | Discussion Paper | Discussion Paper

**ACPD**

doi:10.5194/acp-2015-710

**Typical synoptic situations and their impacts on air pollution**

N. Bei et al.

**Table 2.** Days and percentage of the six types of synoptic situations influencing the Guanzhong basin during the wintertime from 2008 to 2012.

| Categories | North low | Southwest trough | Southeast high | Transition | Southeast trough | Inland high |
|---|---|---|---|---|---|---|
| 2008 | 14 | 8 | 2 | 15 | 11 | 40 |
| 2009 | 17 | 6 | 6 | 14 | 18 | 29 |
| 2010 | 14 | 10 | 8 | 8 | 6 | 41 |
| 2011 | 16 | 6 | 10 | 5 | 3 | 52 |
| 2012 | 22 | 9 | 1 | 15 | 13 | 31 |
| Sum | 83 | 39 | 27 | 57 | 51 | 193 |
| Percentage (%) | 18.4 | 8.7 | 6.0 | 12.7 | 11.3 | 42.9 |

## ACPD

doi:10.5194/acp-2015-710

**Typical synoptic situations and their impacts on air pollution**

N. Bei et al.

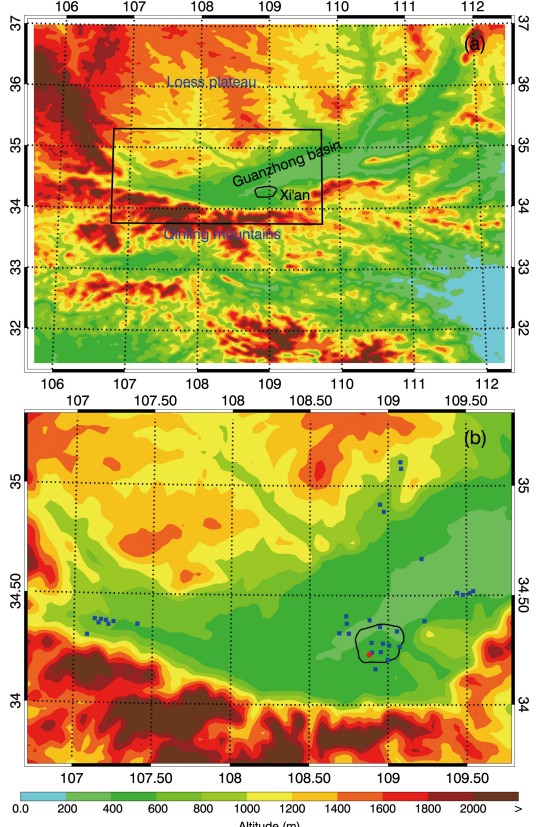

**Figure 1. (a)** WRF and WRF-CHEM model simulation domain with topography and **(b)** geographic distributions of ambient monitoring stations. In **(b)**, the blue filled squares are the ambient monitoring sites and the red filled circle is the IEECAS site.

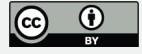



**ACPD**

doi:10.5194/acp-2015-710

**Typical synoptic situations and their impacts on air pollution**

N. Bei et al.



**Figure 2.** Distributions of winds and geopotential heights on 850 hPa at 08:00 BJT on **(a)** 16 February 2014 ("north-low"), **(b)** 19 January 2014 ("southwest-trough"), **(c)** 26 December 2013 ("southeast-high"), **(d)** 2 December 2013 ("transition"), **(e)** 23 January 2014 ("southeast-trough"), and **(f)** 23 December 2013 ("inland-high"). The red filled circle is Xi'an.

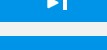
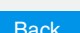


**Figure 3.** 24 h plume transport patterns initialized from 04:00 BJT on **(a)** 16 February 2014 ("north-low"), **(b)** 19 January 2014 ("southwest-trough"), **(c)** 26 December 2013 ("southeast-high"), **(d)** 2 December 2013 ("transition"), **(e)** 23 January 2014 ("southeast-trough"), and **(f)** 23 December 2013 ("inland-high").

**ACPD**

doi:10.5194/acp-2015-710

**Typical synoptic situations and their impacts on air pollution**

N. Bei et al.

**ACPD**

doi:10.5194/acp-2015-710

**Typical synoptic situations and their impacts on air pollution**

N. Bei et al.

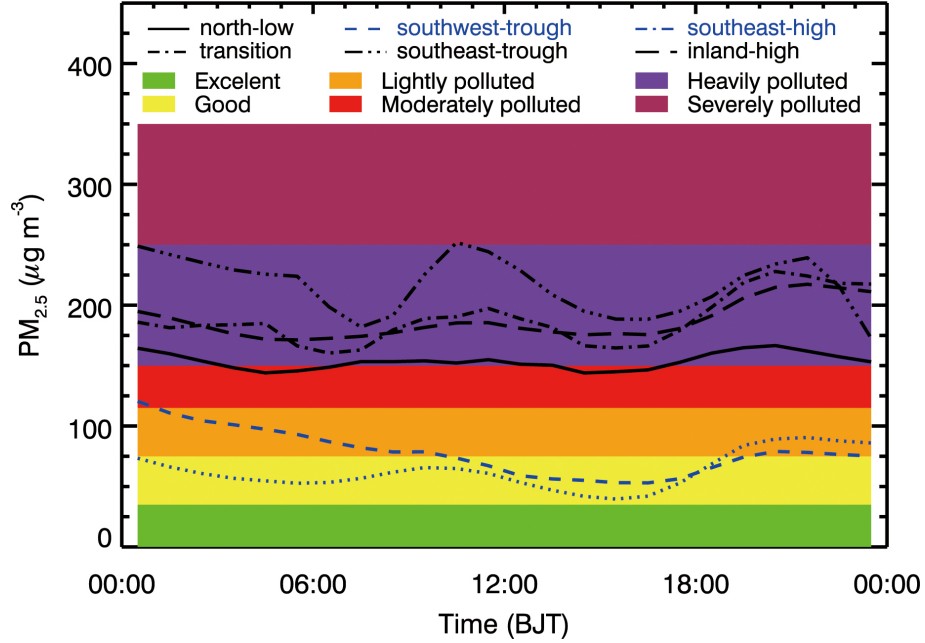

**Figure 4.** Diurnal cycle of observed PM$_{2.5}$ mass concentrations averaged over 33 monitoring sites in the Guanzhong basin under the six synoptic categories during the wintertime of 2013.

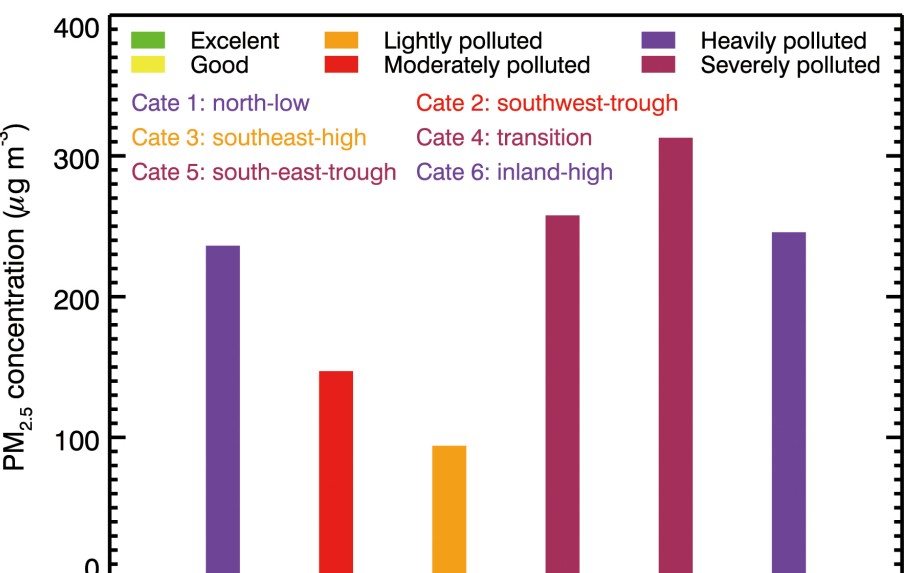

**Figure 5.** Daily mean PM$_{2.5}$ mass concentration averaged during the six typical synoptic situations from 2008 to 2012 at the IEECAS site.

**ACPD**

doi:10.5194/acp-2015-710

Typical synoptic situations and their impacts on air pollution

N. Bei et al.



**ACPD**

doi:10.5194/acp-2015-710

**Typical synoptic situations and their impacts on air pollution**

N. Bei et al.

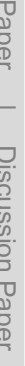

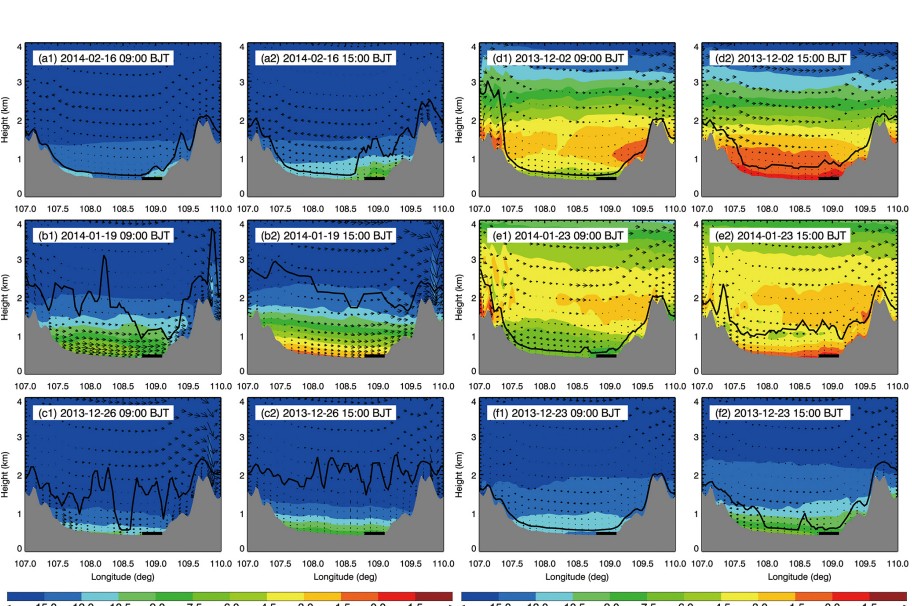

**Figure 6.** Vertical distributions of temperature, wind vectors, and PBL height through Xi'An along the east–west direction at 09:00 and 15:00 BJT on **(a)** 16 February 2014 ("north-low"), **(b)** 19 January 2014 ("southwest-trough"), **(c)** 26 December 2013 ("southeast-high"), **(d)** 2 December 2013 ("transition"), **(e)** 23 January 2014 ("southeast-trough"), and **(f)** 23 December 2013 ("inland-high"). The black filled rectangle represents the urban area of Xi'an, China.

Discussion Paper | Discussion Paper | Discussion Paper | Discussion Paper

**ACPD**

doi:10.5194/acp-2015-710

**Typical synoptic situations and their impacts on air pollution**

N. Bei et al.

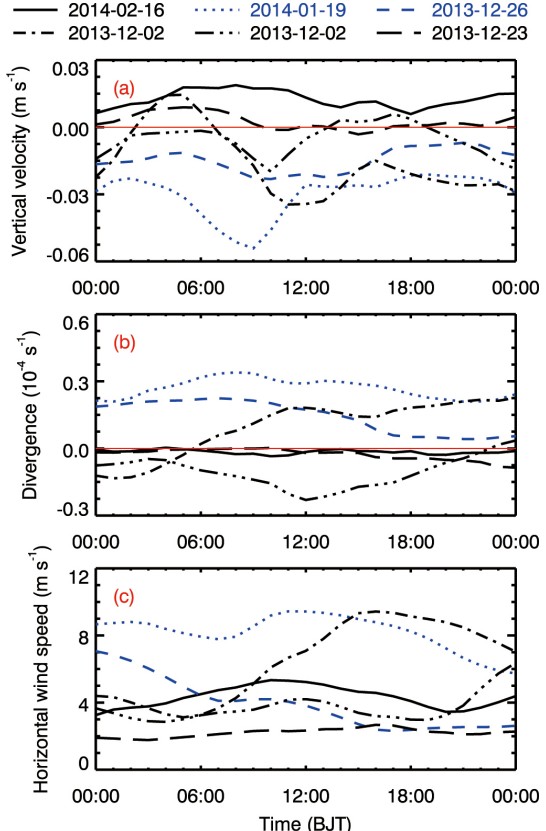

**Figure 7.** Temporal variations of the area averaged low-level **(a)** vertical velocity, **(b)** divergence, and **(c)** horizontal wind speed over the basin.



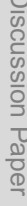

# ACPD

doi:10.5194/acp-2015-710

**Typical synoptic situations and their impacts on air pollution**

N. Bei et al.

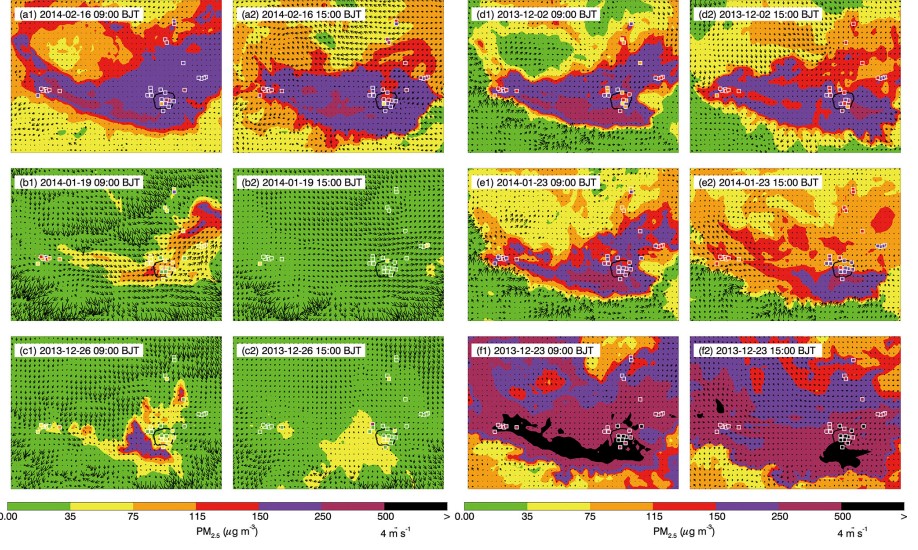

**Figure 8.** Pattern comparison of simulated vs. observed near-surface PM$_{2.5}$ mass concentrations at 09:00 and 15:00 BJT on **(a)** 16 February 2014 ("north-low"), **(b)** 19 January 2014 ("southwest-trough"), **(c)** 26 December 2013 ("southeast-high"), **(d)** 2 December 2013 ("transition"), **(e)** 23 January 2014 ("southeast-trough"), and **(f)** 23 December 2013 ("inland-high"). Colored squares: PM$_{2.5}$ observations; color contour: PM$_{2.5}$ simulations; black arrows: simulated surface winds.

Discussion Paper | Discussion Paper | Discussion Paper | Discussion Paper

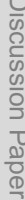

**ACPD**

doi:10.5194/acp-2015-710

**Typical synoptic situations and their impacts on air pollution**

N. Bei et al.

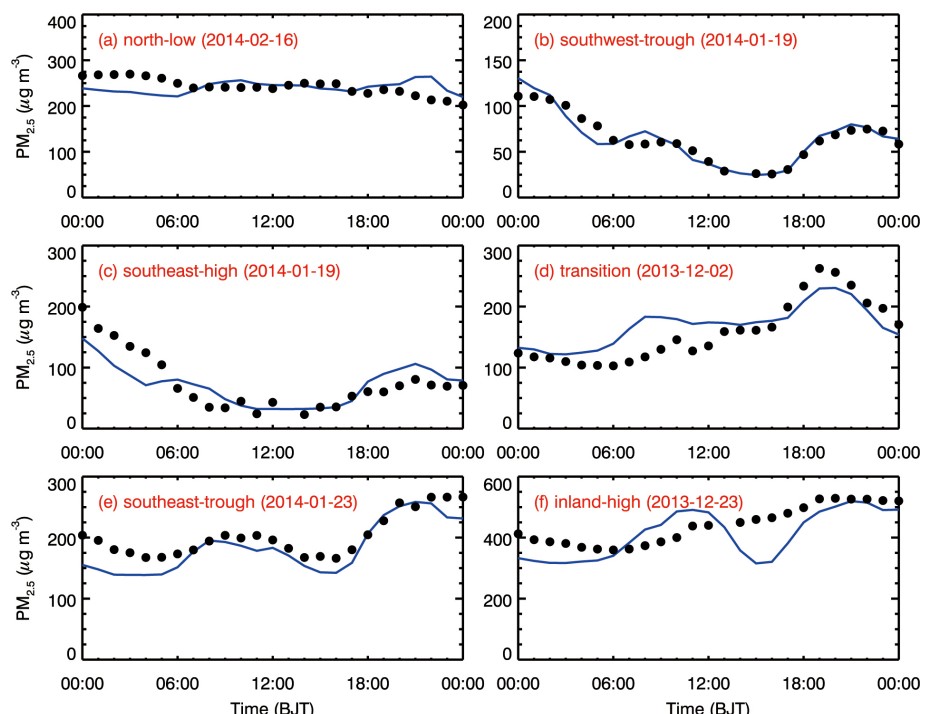

**Figure 9.** Comparison of observed and predicted diurnal profiles of the PM$_{2.5}$ mass concentrations averaged over the monitoring sites in the Guanzhong basin on **(a)** 16 February 2014 ("north-low"), **(b)** 19 January 2014 ("southwest-trough"), **(c)** 26 December 2013 ("southeast-high"), **(d)** 2 December 2013 ("transition"), **(e)** 23 January 2014 ("southeast-trough"), and **(f)** 23 December 2013 ("inland-high").

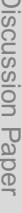

**ACPD**

doi:10.5194/acp-2015-710

**Typical synoptic situations and their impacts on air pollution**

N. Bei et al.

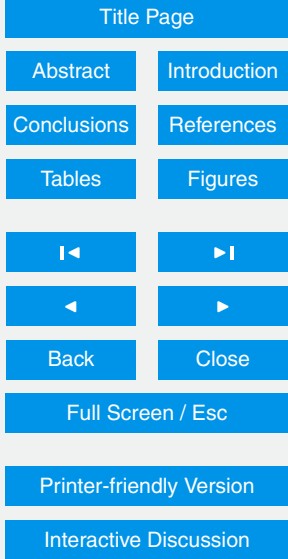

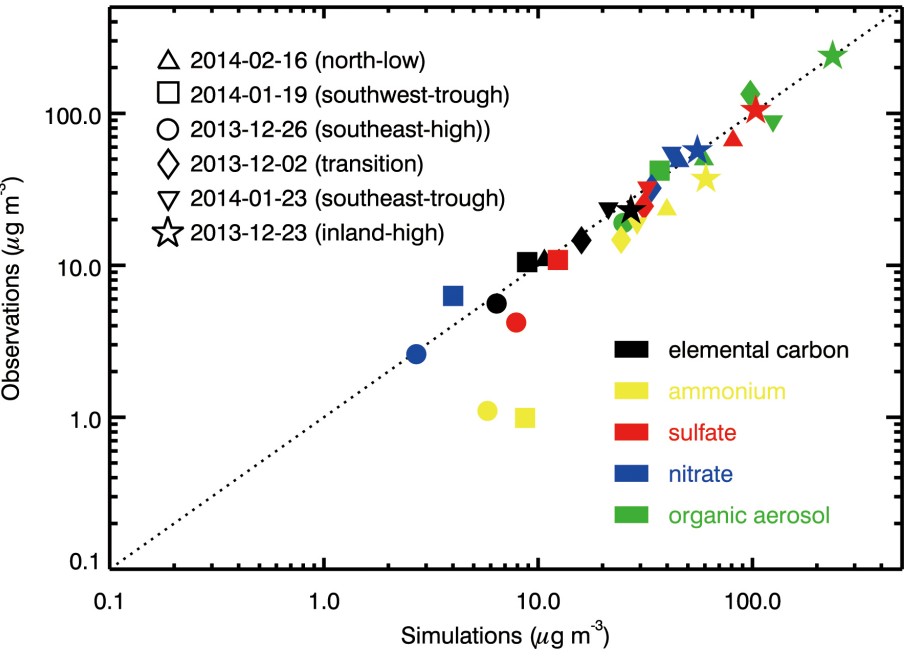

**Figure 10.** Scatter plot of the measured vs. modeled daily mean mass concentration of aerosol constituents at IEECAS site on the selected six days.