# Peer review of "Typical synoptic situations and their impacts on the wintertime air pollution in the Guanzhong basin, China"

_Atmospheric Chemistry and Physics, 2015_

## Referee Comment (RC1) · Anonymous Referee #1 · 19 Feb 2016

This paper develops a meteorological classification for the Guanzhong basin to be used for analyzing air pollution. A subjective method is used based on meteorological reanalysis data. The classification is shown to correspond to different levels of PM2.5 in the basin. The WRF-Chem model is used to evaluate in greater detail an example day for each of the six categories developed in the classification. Overall the paper is clear and the results are relevant, and hence publication is recommended.

Major comments:

It would be helpful to have a bit more description of the subjective procedure used for the classification. Composites could be shown in supplementary material, or some other figure indicating how well defined each group is.

[Figure]

Line 309 and Fig. 7: There should be a better description of what is done for the divergence and vertical velocity. In particular, divergence is usually a sign of suppressed mixing. However, for 1/19 and 12/26 we see both divergence and high mixing heights. The text then claims that the divergence leads to "outflowing of pollutants." Is this not a more straightforward case of strong winds from the north blowing pollutants over the mountain towards the south (rather than towards the east along the basin)? This seemed like a weak part of the paper. I would recommend removing it and including a figure focused on horizontal wind speed and vertical mixing height instead.

There is a brief allusion to dust when discussing Fig 10, but no other mention. It seems that there should be some more discussion of this including some references about dust transport from the north.

The results section is at times confusing and hard to follow, I would recommend some more editing.

Minor comments:

The figure captions should be expanded and be more descriptive. Eg. what is the orange area in Fig. 2? Fig. 6: The transect could be shown in Fig. 1? There is no reference for MOZART. Line 217: This seemed confusing: the "southeast-High" has a high in the "northwest"? The paper needs some further proof-reading and language editing. Examples: "popular" instead of "frequent." Line 395-397 is not clear. "frequently attacks the basin" is a bit dramatic.
* * *

---

## Referee Comment (RC2) · Anonymous Referee #2 · 16 Apr 2016

The manuscript presents a quite comprehensive analysis of the typical synoptic situations influencing the Guanzhong basin. The pollutant transport patterns, meteorological conditions, and observed PM levels in various synoptic situations were discussed in detail. It helps the scientists and policy makers to better understand the underlying reasons for heavy air pollution in this area and might also serve as a statistical basis for air quality forecasting work. Publication is recommended after a few revision comments are addressed.

Specific comments Page 8, Lines 11–15 Please consider briefly showing the analysis procedures in supplementary material.

Page 6, Lines 25–26 Page 8, Lines 19–23 It is stated that six selected days, representing six categorized typical synoptic simulations of the Guanzhong basin, were simulated by the numerical model. Please elaborate a bit about the selection process since the synoptic situations are not very clear cut at times, or even for different days which were grouped into the same category, the PM behavior could be quite different. Did the authors simulate a few cases for every category and then make the selection? Would the model give similar simulation results for most of the cases in the same category?

Figure 8 Seen from the figures, the model simulations tend to underestimate the PM concentrations when the concentration levels are high. What are the author's views on this? What are the major uncertainties of the model?

[Figure]

---

## Author Comment (AC1) · 27 May 2016

Reply to Anonymous Referee #1

We thank the reviewer for the careful reading of the manuscript and helpful comments. We have revised the manuscript following the suggestion, as described below.

Major comments:

Comment: It would be helpful to have a bit more description of the subjective procedure used for the classification. Composites could be shown in supplementary material, or some other figure indicating how well defined each group is.

Response: We have added a paragraph in Section 2 to provide a detailed description of the subjective procedure used for the classification:

"*2.3 Classification Method*

*The subjective procedure is used to categorize the synoptic situations that affect the plume transport patterns in the Guanzhong basin. The synoptic weather system is first identified according to the geopotential height and wind fields on 850 hPa. Then the detailed position of the basin to the weather system can be determined. For example, if the basin is located in the southwest of a trough, the synoptic situation is categorized as "southwest-trough"; and if a high-pressure system controls the basin, the synoptic situation is defined as "inland-high". However, since the synoptic situations are not very clear-cut at times, the FLEXPART-WRF model is further used to calculate the plume transport patterns in the basin under different synoptic situation classifications. If there exists the transition of the weather system influencing the basin for one day, the synoptic categorization is determined by the plume transport patterns in the basin. For example, on some day, the weather system influencing the basin transits from "inland-high" to "southwest-trough". The calm and stable situations induced by "inland-high" facilitate the pollutants accumulation in the basin, but the dry and cold northwest winds caused by "southwest-trough" is subject to evacuate the pollutants in the basin. If the FLEXPART-WRF model results show that the plume moves outside of the basin, the synoptic situation is categorized as "southwest-trough" for the day, otherwise it is classified as "inland-high". Additionally, the occurrence of precipitation is not considered yet in the categorization, which can efficiently wash out pollutants in the atmosphere. Therefore, it is worth noting that, on different days which are grouped into the same category, the pollutants behavior might be quite different caused by the weather system transition or precipitation occurrence.*"

Comment: Line 309 and Fig. 7: There should be a better description of what is done for the divergence and vertical velocity. In particular, divergence is usually a sign of suppressed mixing. However, for 1/19 and 12/26 we see both divergence and high mixing heights. The text then claims that the divergence leads to "outflowing of pollutants." Is this not a more straightforward case of strong winds from the north blowing pollutants over the mountain towards the south (rather than towards the east along the basin)? This seemed like a weak part of the paper. I would recommend removing it and including a figure focused on horizontal wind speed and vertical mixing height instead.

Response: We agree with the reviewer's suggestion, and have removed the old Fig. 7 and included a figure (Fig. 8 now) focused on the horizontal wind speed and vertical mixing height. We have clarified in Section 3:

" *In order to investigate the detailed local meteorological conditions over the Guanzhong basin during the above-mentioned six days, we have further analyzed the low-level (below 850 hPa) horizontal wind speed and PBL height averaged over the Guanzhong basin (the averaged domain indicated in Figure 1). Figure 8 shows the time-evolutions of the area averaged low-level wind speed and PBL height over the basin on the selected 6 days. In general, under favorable synoptic situations, the occurrence of strong horizontal winds speeds up the evacuation of pollutants, such as on Jan. 19, 2014, the average wind speed is around 8 m s$^{-1}$. In addition, the daily average PBL height exceeds 500 m, facilitating the pollutants dispersion in the vertical direction. However, under the unfavorable conditions, except on Dec. 2, 2013, the weak horizontal winds inefficiently disperse the pollutants, i.e., the horizontal wind speed is about 2 m s$^{-1}$ on Dec. 23, 2013. The low PBL also suppresses the vertical dispersion in the basin.*"

Comment: There is a brief allusion to dust when discussing Fig 10, but no other mention. It seems that there should be some more discussion of this including some references about dust transport from the north.

Response: We have included several references and included a figure (Figure 11b) to discuss the dust transport. We have clarified in Section 3:
"*The filter measurements show that the PM$_{2.5}$ in the basin contains abundant potassium, sodium, and calcium ions which are originated from biomass burning or dust and are able to preferentially replace the ammonium ions (Shen et al., 2009, 2011). Figure 10b shows the scatter plot of the elemental iron with potassium measured by filter samples at the IEECAS site on the selected 6 days. In general, the elemental iron is primarily originated from dust and the potassium is mainly attributed by biomass burning. Under unfavorable conditions, the potassium concentration exceeds 3 μg m$^{-3}$, showing considerable amount of biomass burning. Except on Feb. 16, the elemental iron concentration is also high, ranging from 0.5 to 1.5 μg m$^{-3}$, likely caused by the local production or long-range transport from Loess Plateau (Long et al., 2015).*"

Comment: The results section is at times confusing and hard to follow, I would recommend some more editing.

Response: We have revised the manuscript carefully and corrected the errors as suggested. In addition, our co-author Dr. Luisa T. Molina, has edited the grammar carefully.

Minor comments:

The figure captions should be expanded and be more descriptive. Eg. what is the orange area in Fig. 2? Fig. 6: The transect could be shown in Fig. 1? There is no reference for MOZART. Line 217: This seemed confusing: the "southeast-High" has a high in the "northwest"? The paper needs some further proof-reading and language editing. Examples: "popular" instead of "frequent." Line 395-397 is not clear. "frequently attacks the basin" is a bit dramatic.

1) We have expanded the figure captions as the reviewer suggested.

Figure 1. (a) WRF and WRF-CHEM model simulation domain with topography and (b) geographic distributions of ambient monitoring stations. *In (b), the blue filled squares are the ambient monitoring sites, the red filled circle is the IEECAS site, and the cross line is the position of the cross-section shown in Figure 7. The color contour in both panels denotes the terrain height.*

*Figure 2. Composite distributions of winds and geopotential heights on 850hPa at 08:00 BJT for the categories of (a) "north-low", (b) "southwest-trough", (c) "southeast-high", (d) "transition", (e) "southeast- trough", and (f) "inland-high". The red filled circle is Xi'an. The orange shading represents the terrain height over 1500 m.*

Figure 3. Distributions of winds and geopotential heights on 850hPa at 08:00 BJT on (a) 16 February 2014 ("north-low"), (b) 19 January 2014 ("southwest-trough"), (c) 26 December 2013 ("southeast-high"), (d) 2 December 2013 ("transition"), (e) 23 January 2014 ("southeast- trough"), and (f) 23 December 2013 ("inland-high"). The red filled circle is Xi'an. *The orange shading represents the terrain height over 1500 m.*

Figure 4. 24h plume transport patterns initialized from 04:00 BJT on (a) 16 February 2014 ("north-low"), (b) 19 January 2014 ("southwest-trough"), (c) 26 December 2013 ("southeast-high"), (d) 2 December 2013 ("transition"), (e) 23 January 2014 ("southeast-trough"), and (f) 23 December 2013 ("inland-high"). *The gray contour denotes the terrain height. The color dots represent the released particles at different time.*

Figure 7. Vertical distributions of temperature, wind vectors, and PBL height along the cross line *denoted in Figure 1b* at 09:00 and 15:00 BJT on (a) 16 February 2014 ("north-low"), (b) 19 January 2014 ("southwest-trough"), (c) 26 December 2013 ("southeast-high"), (d) 2 December 2013 ("transition"), (e) 23 January 2014 ("southeast-trough"), and (f) 23 December 2013 ("inland-high"). The black filled rectangle represents the urban area of Xi'an, China.

Figure 8. *Temporal variations of the area averaged low-level (a) wind speeds and (b) PBL height* over the basin on (a) Feb. 16, 2014 ("north-low"), (b) Jan. 19, 2014 ("southwest-trough"), (c) Dec. 26, 2013 ("southeast-high"), (d) Dec. 2, 2013 ("transition"), (e) Jan. 23, 2014 ("southeast-trough"), and (f) Dec. 23, 2013 ("inland-high").

Figure 11. Scatter plot of (a) the measured vs. modeled daily mean mass concentration of aerosol constituents and (b) *the elemental iron with potassium measured by filter samples* at IEECAS site on the selected six days.

2) The transect line has been shown in Figure 1b.

3) The reference for MOZART has been included in Section 2:

The chemical initial and boundary conditions for the WRF-CHEM model simulations are interpolated from the 6-h output of a global chemical transport model for $O_3$ and related chemical tracers (MOZART) *(Horowitz et al., 2003)*.

Horowitz, L. W., Waters, S., Mauzerall, D. L., Emmons, L. K., Rasch, P. J., Tie, X., Lamarque, J.-F., Schultz, M. G., Tyndall, G. S., Orlando, J. J., and Brasseur, G. P.: A global simulation of tropospheric ozone and related tracers: Description and evaluation of MOZART, version 2, J. Geophys. Res., 108, 4784, doi:10.1029/2002JD002853, 2003.

4) We have rephrased the sentence in Section 3: *"For "southeast-high" (Figure 3c), the basin is located in the southeast of the high at 850 hPa, which originates the prevalent northeasterly or northwesterly winds, transporting the pollutants outside of the basin and remarkably improving the air quality."*

5) We have revised the manuscript carefully and corrected the errors as suggested. In addition, our co-author Dr. Luisa T. Molina, has edited the grammar carefully. We have changed "popular" to "frequent" in Section 4.

6) We have rephrased the sentence in Section 4: *"In association with the $PM_{2.5}$ measurements released by China MEP, the low $PM_{2.5}$ level or good air quality generally correspond to the favorable synoptic situations and vice versa in the basin."*

7) We have changed "frequently attacks the basin" to "frequently engulf the basin" in Section 1.

---

## Author Comment (AC2) · 27 May 2016

**Reply to Anonymous Referee #2**

We thank the reviewer for the careful reading of the manuscript and helpful comments. We have revised the manuscript following the suggestion, as described below.

Specific comments:

Comment: Page 8, Lines 11–15 Please consider briefly showing the analysis procedures in supplementary material.

Response: We have added a paragraph in Section 2 to provide a detailed description of the subjective procedure used for the classification:

"*2.3   Classification Method*
       *The subjective procedure is used to categorize the synoptic situations that affect the plume transport patterns in the Guanzhong basin. The synoptic weather system is first identified according to the geopotential height and wind fields on 850 hPa. Then the detailed position of the basin to the weather system can be determined. For example, if the basin is located in the southwest of a trough, the synoptic situation is categorized as "southwest-trough"; and if a high-pressure system controls the basin, the synoptic situation is defined as "inland-high". However, since the synoptic situations are not very clear-cut at times, the FLEXPART-WRF model is further used to calculate the plume transport patterns in the basin under different synoptic situation classifications. If there exists the transition of the weather system influencing the basin for one day, the synoptic categorization is determined by the plume transport patterns in the basin. For example, on some day, the weather system influencing the basin transits from "inland-high" to "southwest-trough". The calm and stable situations induced by "inland-high" facilitate the pollutants accumulation in the basin, but the dry and cold northwest winds caused by "southwest-trough" is subject to evacuate the pollutants in the basin. If the FLEXPART-WRF model results show that the plume moves outside of the basin, the synoptic situation is categorized as "southwest-trough" for the day, otherwise it is classified as "inland-high". Additionally, the occurrence of precipitation is not considered yet in the categorization, which can efficiently wash out pollutants in the atmosphere. Therefore, it is worth noting that, on different days which are grouped into the same category, the pollutants behavior might be quite different caused by the weather system transition or precipitation occurrence.*"

Comment: Page 6, Lines 25–26 Page 8, Lines 19–23 It is stated that six selected days, representing six categorized typical synoptic simulations of the Guanzhong basin, were simulated by the numerical model. Please elaborate a bit about the selection process since the synoptic situations are not very clear cut at times, or even for different days which were grouped into the same category, the PM behavior could be quite different. Did the authors simulate a few cases for every category and then make the selection? Would the model give similar simulation results for most of the cases in the same category?

Response: We have selected 3 cases for every category and then made the selection. The FLEXPART-WRF and WRF-CHEM models give similar simulation results for the three cases in the same category generally. We have clarified in Section 3:

"       *For discussion convenience, the following six days are selected to represent the above*

*six typical synoptic situations: (1) Feb. 16, 2014 ("north-low"), (2) Jan. 19, 2014 ("southwest-trough"), (3) Dec. 26, 2013 ("southeast-high"), (4) Dec. 2, 2013 ("transition"), (5) Jan. 23, 2014 ("southeast-trough"), and (6) Dec. 23, 2013 ("inland-high"). For the selection process, three days are first chosen for every category. The FLEXPART-WRF and WRF-CHEM models are then used to simulate the pollutants transport pattern and PM$_{2.5}$ variations and distributions on the three selected days in each category. In general, the simulation results from the two models are similar in the same category, but uncertainties still exist, caused by the weather system transition or occurrence of precipitation. Finally, the most typical day for each category is selected for further analysis and model simulations. The synoptic patterns of the selected six days, shown in Figure 3, are similar to those in Figure 2."*

Please also reference Section 2.3 for the classification method.

Comment: Figure 8 Seen from the figures, the model simulations tend to underestimate the PM concentrations when the concentration levels are high. What are the author's views on this? What are the major uncertainties of the model?

Response: We have clarified in Section 3: "*The WRF-CHEM model generally captures well the observed diurnal variations of the PM$_{2.5}$ mass concentrations, but the model simulations tend to underestimate the PM$_{2.5}$ concentrations when the levels are high. The model biases are mainly from the uncertainties of anthropogenic emissions and meteorological field simulations. The model often underestimates the observed PM$_{2.5}$ mass concentrations during nighttime, which perhaps is caused by illegal emissions that are not reflected in the available emission inventories. In addition, in the afternoon on Dec. 23, 2013, the model considerably underestimates the observation. According to Figure 9f, apparently, the simulated northeast winds are subject to pushing the plume to the south of the basin, causing the model underestimation compared to measurements.*"